# An Up-To-Date Review Regarding Cutaneous Benefits of *Origanum vulgare* L. Essential Oil

**DOI:** 10.3390/antibiotics11050549

**Published:** 2022-04-20

**Authors:** Larisa Bora, Stefana Avram, Ioana Zinuca Pavel, Delia Muntean, Sergio Liga, Valentina Buda, Daniela Gurgus, Corina Danciu

**Affiliations:** 1Department of Pharmacognosy, “Victor Babes” University of Medicine and Pharmacy, Eftimie Murgu Square, No. 2, 300041 Timisoara, Romania; larisa.bora@umft.ro (L.B.); stefana.avram@umft.ro (S.A.); ioanaz.pavel@umft.ro (I.Z.P.); corina.danciu@umft.ro (C.D.); 2Research Center for Pharmaco-Toxicological Evaluation, “Victor Babes” University of Medicine and Pharmacy, Eftimie Murgu Square, No. 2, 300041 Timisoara, Romania; buda.valentina@umft.ro; 3Department of Microbiology, “Victor Babes” University of Medicine and Pharmacy, Eftimie Murgu Square, No. 2, 300041 Timisoara, Romania; 4Multidisciplinary Research Center on Antimicrobial Resistance, “Victor Babes” University of Medicine and Pharmacy, Eftimie Murgu Square, No. 2, 300041 Timisoara, Romania; 5Faculty of Pharmacy, “Victor Babes” University of Medicine and Pharmacy, Eftimie Murgu Square, No. 2, 300041 Timisoara, Romania; sergio.liga96@gmail.com; 6Discipline of Clinical Pharmacy, Communication in Pharmacy and Pharmaceutical Care, “Victor Babes” University of Medicine and Pharmacy, Eftimie Murgu Square, No. 2, 300041 Timisoara, Romania; 7Department of Balneology, Medical Recovery and Rheumatology, Family Discipline, Center for Preventive Medicine, “Victor Babes” University of Medicine and Pharmacy, 300041 Timisoara, Romania; gurgus.daniela@umft.ro

**Keywords:** *Origanum vulgare* L. essential oil, skin, cosmeceutical, pharmaceutical formulations

## Abstract

Due to the plethora of pharmacological activities reported in the literature, *Origanum vulgare* L. is a valuable aromatic plant for the medicine of the XXI century. Recent studies highlight that *Origanum vulgare* L. essential oil (OvEo) has gained attention in the dermatological field due to the cosmeceutical potential correlated with the presence of thymol and carvacrol. As a result of the fulminant expansion of bacterial resistance to antibiotics and the aggressiveness of skin infections, OvEo was extensively studied for its antimicrobial activity against *Staphyloccocus* spp. and *Pseudomonas aeruginosa*. Moreover, researchers have also assessed the anti-inflammatory activity of OvEo, suggesting its tissue remodeling and wound healing potential. Whereas OvEo comprises important biological activities that are used in a wide range of pathologies, recently, essential oils have shown great potential in the development of new therapeutic alternatives for skin disorders, such as acne, wounds or aging. Furthermore, substantial efforts have been committed to the development of modern formulations, such as microemulsions and nanoemulsions, in order to create the possibility for topical application. The review brings to the fore the most recent findings in the dermatological field regarding potential plant-based therapies involving OvEo, emphasizing the modern pharmaceutical formulation approaches and the cutaneous benefits in skin disorders.

## 1. Introduction

Currently, the use of medicinal plants in the form of various extracts (total, concentrated or standardized) and in the form of isolated pure phytocompounds for therapeutic purposes for acute and/or chronic pathologies, as well as for prevention, plays an important role in modern medicine. Thus, in recent decades, the development of phytotherapy has undergone an impressive evolution, especially in the design and implementation of new therapeutic strategies [1]. At present, many of the drugs currently used in therapeutic protocols are obtained from medicinal plants. Moreover, chemical synthetized molecules often have natural molecules as a starting point [2]. It is well known that nowadays the industry is still in full development; furthermore, it is faced with important problems, such as the depletion of resources, as well as waste and pollutant production in large quantities [3]. Attempts are being made to find perspectives such as the use of renewable natural resources accompanied by technologies that lead to a dispersion of pollutants in a smaller amount into the atmosphere and waste management [3,4]. Although chemical synthesis is quite widely used from an industrial point of view, considerable and viable alternatives have been found, such as the development of biotechnology and the use of natural compounds in many fields of activity [4,5,6].

From antiquity to present, essential oils have been one of the most commonly employed natural products. Essential oils (EOs) are natural, complex, liquid and volatile compounds obtained as products of secondary metabolism from aromatic plants. They are usually characterized by a strong odor and can be extracted from different vegetal products of aromatic plants, such as roots, wood, leaves, flowers, fruits and whole plants [7]. It is also important to acknowledge that the production of EOs depends on certain factors, including (i) the method of extraction, (ii) drying and storage, (iii) harvesting time, (iv) climatic conditions, (v) crop location, (vi) types of plant species and (vii) the part of the plant employed [7,8,9]. Obtaining EOs, concentration and purification can be performed via different physico-chemical methods, such as (i) pressing, (ii) distillation (hydro-distillation or steam entrainment) and (iii) extraction (solvent extraction, microwave or ultrasound-assisted extraction) [7,9,10]. The choice of a suitable method for obtaining EOs will depend on the final applicability of the obtained extracts. Volatile oils are widely used all over the world and have numerous applications in multiple industries, such as the pharmaceutical, cosmetics, food and perfume industries [7,11]. As a result of the complexity of active compounds, EOs also have a number of disadvantages, such as high volatility, poor solubility in water and thermal and chemical labilities to the action of exogenous agents [7,12]. Thus, for the pharmaceutic and cosmetic industry, the use of EOs in formulations requires the rigorous control of their conditioning and storage conditions [7].

Indeed, due to the complexity of the active compounds, fragrant properties and a plethora of biological effects, EOs can be considered as a resource of natural therapeutic ingredients that can be used in the formulation of a broad range of cosmetic products (creams, emulsions, conditioners, hair repairing shampoos and lotions) [13,14]. Moreover, it is important to avoid their direct application onto the skin because they can induce the occurrence of serious side effects (irritation, darkening of the skin and allergic contact dermatitis) [15,16,17]. It is mandatory that before application onto the skin, EOs are combined with carrier oils (also known as vegetable or fixed oils) in order to make them less toxic to the skin and increase their absorption [18,19]. With all these benefits, unfortunately, the use of EOs still faces some issues, such as (i) instability, caused by sensitivity to oxygen, light and heat; (ii) limited solubility–hydrophobicity; (iii) high volatility, which reduces their bioactivity; and (iv) risk of toxicity [20,21,22]. Therefore, nowadays, scientific efforts are focused on the possibility of using innovative techniques for EO encapsulation due to the benefits of this approach, which include: (i) the enhancement of oxidative stability, thermostability, photostability and biological activity; (ii) target delivery; and (iii) the protection of pharmaceutical and cosmetic products from undesired alterations [22,23]. In recent years, many novel cosmetic delivery systems (vesicle system—liposome, phytosome and ethosome; micro- and nanoparticles; lipid nanoparticles—solid lipid nanoparticles and nanostructured lipid carriers; cyclodextrins; emulsion systems; and fibrous systema—nanofibres and nanotubes) have been successfully developed to protect EOs, to achieve higher solubility and stability, in order to enhance the cellular absorption and bioavailability of cosmetic ingredients [22,23,24,25,26,27,28]. Consumers have growing demands for natural degradable, non-toxic and harmless active compounds, and this awareness has especially intensified the focus of cosmetic companies on natural resources [13,29]. The categories in which essential oils can be found are sorted into a small number of families, such as Lamiaceae, Asteraceae, Apiaceae, Umbellifereae, Myrtaceae, Rutaceae and Poaceae [30,31,32].

In terms of chemical variety, in addition to the primary metabolites found in all vegetal products, there are also secondary metabolites, such as terpenenoids, which are the most common and abundant in EOs [24,33,34]. Terpenoids are obtained from primary metabolic precursors and are generated via various biosynthetic pathways, such as mevalonate and deoxyxylulose phosphate pathways (Figure 1) [24,35,36,37].

*Origanum vulgare* L., commonly known as oregano, belongs to the Lamiaceae family. It is native in Europe and Northern America and cultivated in different parts of the world [38,39,40]. The genus *Origanum* is currently present in various zones of the planet, especially in the Mediterranean region, and includes approximately 45 species, six subspecies and three varieties [41,42]. In therapy, the most frequently used types of extracts are from the aerial plant (leaves, flowers and seeds) [10]. According to the literature, from a chemical point of view, the representative compounds responsible for the main therapeutic activities identified in different OvEos are the isomeric phenolic monoterpenes: carvacrol or 2-methyl-5-(1-methylethyl)-phenol and thymol or 2-isopropyl-5-methylphenol. They are biosynthesized via the aromatization process and hydroxylation of *p*-cymene. However, the chemical composition of OvEos is diverse and includes different compounds, such as acyclic, monocyclic and bicyclic monoterpenes (γ-terpinene, *p*-cymene, linalool, geraniol, β-myrcene, *trans*-sabinene, α-pinene and terpinen-4-ol), sesquiterpenes (1,8-cineole, β-caryophyllene and germacrene-D, β-citronellol) (Figure 2), flavonoids (luteolin, apigenin and quercetin), phenolic acids (rosmarinic and chlorogenic acids) and tannins [42,43,44]. In terms of chemical composition, research on EOs *Origanum vulgare* L. species from different countries has been evaluated and described. A brief presentation of the phytochemical composition of OvEo was previously described by our research group [45]. Different patterns have been reported depending on the pedoclimatic conditions, geographical location and growth conditions [34]. However, the presence of carvacrol and thymol as main representatives is unanimously accepted [42,46].

The phytochemical profile of EOs varies in the number of molecules, chirality and stereochemistry of components and is considerable important, because the spatial orientation of molecules can influence the pharmacological activity of the compound [47]. In the chemistry of EOs, the two main types of key stereoisomers are enantiomers and diastereomers, which are separated using different instrumental methods of chemical analysis (gas chromatography and nuclear magnetic resonance) [47,48]. It has also been reported by several studies that OvEo has a remarkable phytochemical polymorphism with several chemotypes in concordance with the geographical area. Lukas et al. investigated 581 individual plants from 51 populations of European oregano and detected 87 mono- and sesquiterpenes in total. In order to allow a more accurate representation of chemotype distribution, EOs were classified according to metabolic pathway: (i) 175 individual plants were classified into cymyl-compounds (*p*-cymene, carvacrol, thymol, γ-terpinene, *p*-cymene-8-ol, (*E*)- and (*Z*)-β-ocimene, thymol ethyl ether and carvacrol methyl ether); (ii) 239 individual plants into the sabinyl-compounds (sabinene and *trans*-/*cis*- sabinene hydrate); and (iii) 36 plants into in the acyclic (+)-linalool. Sesquiterpenes were also present in greater amounts, such as (*E*)-(−)-β-caryophyllene, (−)-germacrene-D, (*E*,*E*)-β-farnesene, (*E*)-γ-bisabolene, germacrene-D-4-ol and caryophyllene oxide [42]. De Falco et al. analyzed the chemical composition in the different spatial distributions of the plants and showed that EO from plants grown on single rows is rich in sabinene, while that from plants grown in double rows is richer in ocimenes ((*E*)-β-ocimene and (*Z*)-β-ocimene). Additionally, the main constituents present in OvEo are carvacrol and thymol, monoterpenes (β-myrcene, α-terpinene, *p*-cymene and D-(+)-α-pinene), oxygenated monoterpenes (terpinen-4-ol, 1,8-cineole, (*Z*)-sabinyl acetate, *trans*-sabinene hydrate, *cis*-/*trans*-*p*-menth-2-en-1-ol and linalool) and sesquiterpenes ((*E*)-(−)-β-caryophyllene, (−)-germacrene-D, α-humulene, (*E*)-β-farnesene, (*E*,*E*)-α-farnesene and γ-cadinene) [49].

Not only does oregano essential oil have an impressive background in traditional medicine, OvEo is also a natural source for obtaining plant-based drugs. Furthermore, in accordance with the aforementioned, the active phytocompounds that are present in OvEo have many pharmacological benefits, such as antimicrobial, antioxidant, anti-inflammatory, antiproliferative, vasoprotective, antidiabetic, antiaging and wound healing effects [44,46,50,51,52].

Establishing the right concentration of volatile oil is essential in order to avoid the cytotoxic effect of OvEo. Based on this, researchers’ attention is focused on the development of modern technological strategies and the evaluation of cytocompatibility in order to assure the safe usage of pharmaceutical formulations [53,54,55]. Along these lines, García-Salinas et al. conducted a series of cytotoxicity assays on human dermal fibroblasts, human epidermal keratinocytes (HaCaT) and THP1 human monocytes. Researchers concluded that EO compounds such as thymol and carvacrol presented subcytotoxic concentrations in the range 0.015–0.090 mg/mL, while the widely used antiseptic substance, chlorhexidine, appeared to be more cytotoxic than the aforementioned compounds, with a subcytotoxic concentration of 0.004 mg/mL [55]. On the other hand, Spagnoletti et al. determined that *Origanum vulgare* subsp. *hirtum* EO presented an IC_50_ value of 0.148 mg/mL on HaCaT cells [56]. Furthermore, Kapustova et al. demonstrated that the encapsulation process could enhance the cytotoxic effect of OvEo on the HaCaT cell line in a dose-dependent manner. However, nanoencapsulated OvEo inhibited an antibiofilm activity against *Escherichia coli* (*E. coli*) at subcytotoxic concentrations (0.03 mg/mL), while the IC_50_ values obtained for pure and encapsulated oregano oil were 0.093 and 0.044 mg/mL, respectively [57].

In recent years, approaches have been conducted for the design of different pharmaceutical and cosmetic formulations. Recent studies have shown the possibility of incorporating OvEo into topical formulations (nano- or microemulsion) using controlled release systems–nanocarriers obtained via encapsulation techniques [58,59]. From this perspective, the present work aims to provide an up-to-date review on the great potential of OvEo cutaneous applications into the dermatology field according to pharmacological activities. In the following sub-sections, we briefly discuss studies regarding the antimicrobial, antioxidant, anti-inflammatory, antiaging, antiacne and wound healing effects of *Origanum vulgare* L. essential oil.

## 2. Objective and Methodology

The objective of this review is based on the need of a structured collection of information regarding the cutaneous applications of OvEo. Aspects like the potential of OvEo to be used as a natural alternative/complementary approach in acne and wound healing therapeutic protocols due to the antimicrobial and anti-inflammatory activities, as well as the contribution of the antioxidant effect of OvEo in skin aging retardation are gathered together.

The methodology for the design of this review followed the steps already reported by Guiné et al. [60,61]. Subsequently, a search on the PubMed, Web of Science, Science Direct, Mendeley Data and Scopus scientific databases was conducted. The aforementioned databases were systematically searched for articles and the search string combined the keyword with the Boolean operator OR, AND. For instance, the following search string has been employed: ((“origanum vulgare”) OR (“oregano”) AND (“essential oil”) OR (“volatile oil”) AND (“skin”) OR (“cutaneous”) AND (“disease”) OR (“disorder”) AND (“antibacterial”) OR (“antimicrobial”) AND (“inflammation”) AND (“antioxidant”)). The references of the articles of interest were also explored in order to detect other relevant information. To limit the chances of selecting irrelevant or improper data, some inclusion and exclusion criteria were established based on the aspects addressed in the review. Studies assessing other forms of extract than essential oil were automatically excluded. The figures are original and were designed by the help of ServierMedicalArt, BioRender and ChemDraw Ultra 12.0.

## 3. The Benefits of the Antimicrobial Effect on the Skin. The Antimicrobial Activity of *Origanum vulgare* L. Essential Oil

Healthcare professionals have been struggling for years to control the fulminant expansion of bacteria resistance to antibiotics by trying to discover new molecules with potential activity against pathogens such as *Staphylococcus aureus* (*S. aureus*), *Pseudomonas aeruginosa* (*P. aeruginosa*) and *E. coli*. Due to the fact that many natural compounds, such as diterpenes, triterpenes, sesquiterpene lactones, flavonoids and naphthoquinones, possess antimicrobial activity, the study of medicinal plants has gained researchers’ attention with the view to combat bacterial resistance in recent years [62,63,64]. Moreover, the use of plant-based substances for antimicrobial activity in the cosmetic industry can also play a role in maintaining the microbiological purity of a finite product during the validity period [65].

Since the time of Hippocrates, around the 5th century B.C., *Origanum vulgare* L. was utilized for its antimicrobial activity and aid in dermatological, respiratory and digestive infections [66]. Carvacrol and thymol are the two phenols responsible for the antimicrobial capacity of oregano essential oil [67]. Due to their hydrophobic character, the compounds act by dissolving the hydrophobic part of the bacteria membrane, increasing the membrane permeability and causing a loss of the structure in the phospholipid bilayer. In other words, the two terpenes cause structural and functional damage to the cell membrane (Figure 3) [45,68,69]. Our research group published a comprehensive review regarding the bioactivities of OvEo, including the antimicrobial effect [45]. However, due to the specificity of the present review, this section was designed in order to create a full picture regarding the antimicrobial effect of OvEo against cutaneous pathogens involved on skin level.

Most bacterial skin and soft tissue infections occur due to infection with *S. aureus*, a Gram-positive bacterium whose toxins lead to the production of proinflammatory cytokines and mediators and an overall inflammatory response of the organism [70]. Infection with *S. aureus* can affect the superficial layer of the skin with the production of impetigo or infected abrasions, as well as the subcutaneous tissue or the dermis of the skin, leading to complicated skin infections (cellulitis, furunculosis, folliculitis, abscesses, wounds and ulcerations) [71]. Furthermore, methicillin-resistant *S. aureus* (MRSA) strains represent a critical threat for public health due to its resistance to antibiotics and the aggressiveness of the skin infections [70]. Therefore, the antimicrobial activity of OvEo has extensively been explored over time.

In a study published by Guarda et al., the antimicrobial capacity of microencapsulated carvacrol and thymol was determined. The two compounds showed different Minimal Inhibitory Concentration (MIC) values, exhibiting antimicrobial activity against the analyzed microorganisms, including *S. aureus* (thymol–MIC = 0.250 mg/mL; carvacrol–MIC = 0.225 mg/mL). Moreover, researchers determined that thymol and carvacrol can be microencapsulated together for proper antimicrobial activity when using an optimal synergistic ratio [68]. According to Fratini et al., the OvEo (purchased from Pisa, Italy) contained carvacrol (65.93%) as the most representative compound. The research team evaluated the antibacterial activity by microdilution assay against fourteen *S. aureus* strains and obtained remarkable values of MIC (MIC ≤ 0.240 mg/mL for eight strains and MIC ≤ 0.480 mg/mL for five strains) [72]. Grondona et al. applied the disc diffusion technique to assay the antibacterial character of OvEo collected from Córdoba, Argentina against eight bacterial strains, including *S. aureus.* The results suggested that the volatile oil presented maximum activity against the *S. aureus* strain, with a value of antimicrobial activity of 91.20 ± 1.31 AU/mL (arbitrary units per mL calculated as the mean diameter (mm) of the minimal inhibition zone × dilution factor × 50) at a dilution of 1/10 [69]. In order to characterize the antimicrobial activity of nineteen EOs, including *Origanum vulgare* L. volatile oil, Chaftar et al. performed MIC by microdilution assay. OvEo was purchased from Clermont-Ferrand, France. The bacterial and fungal strains commonly involved in the production of skin infections and evaluated in this study included *Staphylococcus epidermidis* (*S. epidermidis*), *P. aeruginosa*, *Trichophyton mentagrophytes* (*T. mentagrophytes*) and *Trichophyton rubrum* (*T. rubrum*). The chemical composition of OvEo (carvacrol—66.89%; thymol—4.65%) is directly responsible for the highly antibacterial and antifungal activities of both Gram-positive (MIC ≤ 1.13 mg/mL) and Gram-negative (MIC ≤ 0.34 mg/mL) bacteria and fungal species (MIC ≤ 1.80 mg/mL). Due to the obtained results, researchers concluded that OvEo may be a promising dermatocosmetic compound [73]. In the same vein, employing MIC assay, Nostro et al. evaluated the sensibility of methicillin-resistant *S. aureus* (MRSA) and methicillin-resistant *S. epidermidis* (MRSE) to commercial OvEo, carvacrol and thymol. Results demonstrated that all bacterial strains were susceptible to the evaluated compounds with no significant differences. Carvacrol showed the best MIC values (0.015–0.03%), closely followed by thymol (0.03–0.06%) and OvEo (0.06–0.125%). Therefore, the findings strongly suggest that OvEo can be used as a potential antibacterial agent against MRS strains in topical formulations [74].

Although skin infections with *P. aeruginosa* are not as common as those with *S. aureus*, this Gram-negative bacteria can be involved in severe forms of dermatosis (ecthyma gangrenosum) due to the colonization of wounds and burns of the hospitalized patients. *P. aeruginosa* may also produce dermatitis or folliculitis favored by activities in improperly disinfected water [75]. The World Health Organization declared this ubiquitous microorganism the number one priority pathogen due to the life threatening nosocomial infections caused and the high level of antibiotic resistance [76]. Bejaoui et al. evaluated the antibacterial activity of Spanish *Origanum vulgare* L. subsp. *glandulosum* at different phenological stages (vegetative, late vegetative and flowering period) by MIC and disc diffusion assays. One of the tested bacteria was *P. aeruginosa* (NCTC 10418 strain), with a value of the inhibition zone of 9 mm and a high value of MIC (0.125 mg/mL) at all phenological stages [77]. The results obtained by Özkalp et al., who studied the antimicrobial effect of OvEo collected from Mersin (Büyükeceli-Gülnar) in Turkey against *P. aeruginosa* (RSKK 06021 strain) and other microorganisms, are also noteworthy. By applying broth microdilution assay, the research team obtained the same MIC value of oregano volatile oil as for ampicillin used as a control compound (MIC = 0.064 mg/mL). In addition, *Origanum vulgare* L. possessed inhibitory activity on all the tested bacteria at a very low concentration of the volatile oil [78]. Complementarily, Soumya et al. evaluated the inhibitive activity of carvacrol and thymol on the *P. aeruginosa* biofilm formation, with results that encourage further studies for antiadherence and antibiofilm natural compounds [79].

## 4. The Benefits of the Antioxidant Effect on the Skin. The Antioxidant Activity of *Origanum vulgare* L. Essential Oil

Skin represents the largest organ of the human body and is well known for its role in protection against various external factors, such as microorganisms, allergens, xenobiotics and ultraviolet irradiation [80,81]. It has been also very well documented the fact that oxidative stress is induced by an exaggerated amount of reactive oxygen species (ROS) and is involved in the production of numerous cutaneous diseases [82]. As is well known, ROS are free radical species of oxygen, characterized by a reactive state, and include hydrogen peroxide (H_2_O_2_), superoxide radical (O_2_•−), hydroxyl radical (HO•) and singlet oxygen (^1^O_2_) [83]. Elements such as pollutants, ultraviolet (UV) radiation, drugs and cosmetic products, growth factors and cytokines stimulate skin cells, especially keratinocytes, to produce ROS. As a response, these molecules can directly deteriorate lipid membranes, collagen, amino acids and DNA structure [84]. The damage caused by ROS can be kept under control by antioxidant compounds. The cutaneous antioxidant system comprises enzymatic and non-enzymatic substances, which are endogenously synthesized (superoxide dismutase, glutathione peroxidase/dismutase, catalase, estradiol, estrogen and melanin) or have an exogenous origin (vitamin C and E, and carotenoids) [85]. Higher levels of antioxidants are found in the top layer of the epidermis, due to the higher O_2_ partial pressure at the surface [86]. These protective substances act by inhibiting the formation of free radicals and must present a lower redox potential than the protected compound, in order to be first oxidized, or by chelating metal ions [87]. Additionally, in the *stratum corneum*, the SPRR protein family (small proline rich repeat proteins) can be found, which acts as a defense mechanism against ROS by forming disulfide bonds between molecules [86]. Therefore, significant changes in the ROS–antioxidant balance are involved in the pathogenesis of many skin diseases, including skin aging, skin cancer, depigmentation (vitiligo) and skin inflammation (acne, atopic dermatitis and psoriasis) [88].

OvEo is intensively studied for its antioxidant activity correlated with the presence of thymol and carvacrol, two terpenes with major contributions in many pharmacological activities of this plant. The mechanisms of action of the antioxidant effect are based on the property of the two terpenes to form complexes with free radicals and to chelate metal ions (Figure 4) [89]. Quiroga et al. studied the relationship between the chemical composition of four species of oregano and the antioxidant activity. The plants were harvested in Argentina, and the experiment included *Origanum vulgare* subsp. *virens* (Hoffm. et Link) letswaart, *Origanum × applii* (Domin) Boros, *Origanum × majoricum* Cambess and *Origanum vulgare* L. subsp. *vulgare.* Employing gas–liquid chromatography and mass spectrometry (GC–MS) to determine the essential oil composition and 2,2-diphenyl-1-picrylhydra-zyl (DPPH) assay to analyze the antioxidant activity, the researchers obtained a higher thymol content for *Origanum vulgare* subsp. *virens* (29.7%) and *Origanum vulgare* subsp. *vulgare* (26.6%) and a radical scavenging activity of 0.98 and 0.90 µg/mL, respectively [90]. In the same manner, Borgarello et al. evaluated the free radical-scavenger activity of OvEo collected in the central area of Argentina by applying the DPPH method. The research group obtained a value of IC_50_ equal to 0.44 mg/mL for the essential oil, which was significantly higher than the values obtained for the fraction with increased antioxidant capacity (R3) and the commercial antioxidant butylated hydroxytoluene (0.22 and 0.20 mg/mL, respectively) [91]. Zheljazkov et al. studied the effect of distillation time on the antioxidant activity of OvEo acquired from Rancho Cordova, CA, employing the oxygen radical absorption capacity (ORAC) method. The authors concluded that even though the time of hydrodistillation was gradually increased (20, 80 and 360 min), the values did not differ substantially in their antioxidant capacity (60.8, 74.5 and 60.4 µmol Trolox equivalents/g, respectively) [92]. Using the same method, Almeida et al. reported that pure oregano essential oil produced from Alentejo, Portugal had an activity of 1610 ± 150 µmol TEAC/g [93]. Sarikurkcu et al. observed the free radical scavenging activity of the Turkish *Origanum vulgare* subsp. *hirtum* and *Origanum vulgare* subsp. *vulgare* using 2,2 azino-bis (3-ethylbenzothiazoline-6-sulfonic acid) radical cation (ABTS) assay and obtained amounts of 176.41 ± 0.22 and 9.63 ± 1.01 mg TEs/g oil, respectively [94]. By applying DPPH, ABTS and ORAC lipophilic assays, Tapiero et al. evaluated the antioxidant capacity of two different samples of OvEo collected from Vijes and Sevilla Valle del Cauca, Colombia for the concentrations of 5, 10 and 40 µg/mL. The DPPH method recorded the highest value of equivalent Trolox, namely, between 310.8 and 320.6 mmol Trolox/100 g, while ABTS and lipophilic ORAC assays registered considerably less activity (23.128–24.019 and 0.3014–0.3256 mmol Trolox/100 g, respectively) [95].

## 5. The Benefits of the Anti-Inflammatory Effect on the Skin. The Anti-Inflammatory Activity of *Origanum vulgare* L. Essential Oil

The permanent interaction of the skin with exogenous factors, the overall daily lifestyle and elements such as skin barrier and immune dysfunctions, microbiota disturbance, gene mutations and neuroinflammation can exacerbate or lead to immune-mediated skin disorders, such as contact dermatitis, atopic dermatitis and psoriasis. All of these skin diseases are characterized by a common symptom: inflammation [96,97,98]. The inflammatory response of the organisms involves the proliferation and infiltration of proinflammatory cytokines such as TNF and IL-23/IL-17 in the pathogenesis of psoriasis, while immune cells such as thymic stromal lymphopoietin (TSLP), IL-33, IL-1β and IL-8 characterize the pathogenesis of contact/atopic dermatitis [98]. In order to overcome the limitations of some anti-inflammatory synthetic compounds or to avoid their side effects, in recent years, researchers have tried to incorporate oregano essential oil into nano- and microemulsions as colloidal drug carriers to create the possibility for topical application [45].

*Origanum vulgare* L. volatile oil has been amply studied for its anti-inflammatory activity due to the presence of carvacrol and its action mechanisms, which include the reduction in TNF-α, IL-1β, IL-6 and IL-8; the induction of IL-10 release; and the inhibition of NADPH oxidase, lipoxygenase and reactive oxygen species (Figure 5) [44,99]. Considering the biological activity of commercially procured OvEo (dōTERRA International LLC, Pleasant Grove, UT, USA), Han et al. conducted an in vitro study in a cell culture of human dermal fibroblast (BioMAP HDF3CGF). The system is designed to imitate fibrosis and chronic inflammation. The GC–MS method attested carvacrol as the major compound of OvEo (78%). ELISA revealed significantly reduced levels of the following inflammatory biomarkers: monokine induced by γ-interferon (MIG), monocyte chemoattractant protein 1 (MCP-1), interferon γ-induced protein 10 (IP-10), interferon-inducible T-cell ɑ-chemoattractant (I-TAC) and vascular and intracellular cell adhesion molecule 1 (VCAM-1/ICAM-1), in a concentration-dependent manner. IL-8 was weakly inhibited. Tissue remodeling biomarkers were also significantly decreased by OvEo. Thus, the promising results obtained in this research paper suggest the anti-inflammatory, tissue remodeling and wound healing potential of oregano essential oil [100]. The evaluation of the anti-inflammatory activity of carvacrol by Silva et al. is also noteworthy. The research group studied the evolution of experimental models of edema induced by different agents in Wistar rats and Swiss mice after treatment with carvacrol (purity 98%), obtained from St. Louis, MO, USA. All the results were compared with NaCl 0.9% used as a vehicle. In the paw edema model induced by histamine or dextran, carvacrol presented a decreasing activity of edema of 46% and 35%, respectively, at a dose of 50 mg/kg, while cyproheptadine decreased the edema formation by 61% and 43%, respectively, at a dose of 10 mg/kg. In accordance with this, in the paw edema model induced by substance P, carvacrol (100 mg/kg) also decreased edema formation (46%). In the ear edema models induced by 12-*O*-tetradecanoyl-phorbol acetate or arachidonic acid, carvacrol (0.1 mg per ear) showed a reduction in the formation of ear edema (43% and 33%, respectively). At the same time, indomethacin (0.5 mg or 2.0 mg per ear) inhibited inflammation by 55% and 57%, respectively. In addition, the research team also evaluated the evolution of gastric lesions induced by acetic acid in rats treated orally with carvacrol (25, 50 and 100 mg/kg) for a period of two weeks. The tested compound significantly reduced the lesion area (60%, 91% and 81%, respectively). Therefore, the team concluded that the obtained results are promising in order to use carvacrol in inflammatory skin diseases, due to the association with edema and leukocyte infiltration on which the terpene acts actively [101]. In the same vein, Lima et al. also evaluated the anti-inflammatory activity of carvacrol (98% purity, purchased from St. Louis, MO, USA) in a paw edema model induced in Swiss mice. ELISA, real-time PCR and enzyme immunoassay revealed significantly decreased levels of IL-1β and PGE2 and the down-regulation of COX-2 expression, as well as increased levels of IL-10. Therefore, the anti-inflammatory property of carvacrol is closely related to its capacity of IL-10 induction [102].

## 6. The Antiaging Effect of *Origanum vulgare* L. Essential Oil

Cutaneous aging is defined as a time-dependent process caused by internal and external factors, which impacts the functionality of the skin. Aging is commonly associated with skin fragility due to the thinned and atrophic epidermis, a decreased production of collagen and elastin and a slowed cell turnover rate, with visible manifestations, such as wrinkles, age spots and dry skin [103]. Withal, collagenase, elastase and hyaluronidase are known for their negative effects against the skin structure and its components (the cleavage of elastin and collagen fibrils and a decrease in hyaluronic acid) [104]. Although there are many reports in the scientific literature regarding the antiaging capacity of natural compounds, especially polyphenols [105], *Origanum vulgare* L. EO has received attention from researchers only in recent years.

The main objective of the research paper conducted by Laothaweerungsawat et al. was the characterization of the relationship between the chemical composition and the antiaging properties of OvEo from two different regions. Therefore, samples of oregano cultivated from a tropical region (Thailand–HO) and commercial oregano obtained from a Mediterranean region (Spain–CO) were used. The two volatile oils had a similar chemical composition, with carvacrol as the main component with a percentage of 79.5% for HO and 64.6% for CO. Furthermore, researchers observed the antiaging capacity of HO and CO by means of anticollagenase, antihyaluronidase and antielastase activity assays. Results showed that HO possessed superior inhibitory activity against collagenase and elastase (92.0 ± 9.7% and 53.1 ± 13.3%, respectively) compared to CO, with IC_50_ values of 35.1 ± 0.9 µg/mL against collagenase and 24.3 ± 0.5 µg/mL against elastase. The volatile oils did not possess any efficient hyaluronidase inhibitory activity (16.7 ± 0.3% and 15.5%, respectively) compared to oleanolic acid used as a positive control (86.0 ± 1.1%). Whereas carvacrol did not present satisfying results regarding the enzyme’s inhibitory activity, the research group concluded that the antiaging activity of OvEo could be due to the synergy of several compounds. The findings of the study suggested the great potential of tropical *Origanum vulgare* L. to be used for its skin aging retardation effect [104]. Similarly, Lee et al. demonstrated that carvacrol possessed an antiwrinkle effect via the induction of collagen production in human dermal fibroblasts, sustaining the possibility of using this compound in skin aging retardation [106].

It is well known that chronic exposure to UVB rays is strongly associated with photoaging and wrinkle formation, involving decreased levels of collagen due to an abnormal synthesis and irregular collagen bundle formation. Along these lines, Ili et al. studied the effects of Turkish *Origanum hypericifolium* O. Schwartz and P.H. Davis essential oil on UVB-irradiated skin of female BALB/c mice. They showed that the main components of oregano volatile oil were cymene (34.33%), carvacrol (21.76%) and thymol (19.54%). The histological examination revealed the fact that the group exposed to UVB irradiation and treated with essential oil did not possess a substantial loss of collagen, degraded collagen bundles or a remarkable accumulation of elastic fibers. Therefore, the findings suggest that *Origanum hypericifolium* O. Schwartz and P.H. Davis essential oil could ensure a protective capacity against the alterations of collagen and elastic fibers induced by UVB rays, mainly due to the carvacrol content [107].

Due to the fact that hyperpigmentation represents visible and unpleasant signs of skin aging, El Khoury et al. evaluated the antimelanogenic activity of *Origanum ehrenbergii* Boiss. and *Origanum syriacum* L. essential oils (obtained from Abbadiye (Shouf) and Ebrine (Batroun), Lebanon) and carvacrol as the major compound of these oils in vitro on B16-F1 murine melanocytes. The tyrosinase and melanin levels in melanocytes were measured after the treatment with the two essential oils with pure carvacrol, respectively. Experiments have shown that the two essential oils and carvacrol showed a significant decrease in melanin levels (15–20% and 30%, respectively), while the levels of tyrosinase remained unchanged. The obtained results demonstrate the capacity of the two *Origanum* species to block melanogenesis, suggesting the usefulness in cutaneous hyperpigmentation diseases that occur with aging [108].

A snapshot of the antiaging activity of OvEo is presented in Figure 6.

## 7. The Antiacne Effect of *Origanum vulgare* L. Essential Oil

Acne is a common skin condition, which affects mostly teenagers and is associated with a negative mental impact and an overall decreased quality of life. Inflammation, increased sebum production, hyperkeratinization and microbial infection represent the interconnected mechanisms involved in the pathogenesis of this disease. The representative microorganisms isolated from acne lesions are *Propionibacterium acnes* (*P. acnes*), *S. epidermidis* and *S. aureus*. Due to the fact that the treatment compliance of the current antiacne therapies can be poor, as a result of factors such as side effects or discontinuous administration, the introduction of natural treatments in acne therapeutic protocols may be a suitable future approach [109,110].

In order to overcome the limitations of topical antiacne antibiotics, Taleb et al. investigated seven essential oils with potential antiacne activity and developed a pharmaceutical formulation of the essential oil that possessed the highest antibacterial effect. Due to the fact that OvEo (99.4% thymol), which was provided from Lincoln, NE, USA, exhibited the strongest antimicrobial activity, the volatile oil was selected to be formulated as a nanoemulsion and to be tested in vivo for its topical antiacne activity. Therefore, OvEo was topically applied to infected male BALB/c mice ears with *P. acnes* in a dose of 0.672 mg/mL. A second group of mice was treated with 2% erythromycin epicutaneously applied (positive control). The OvEo formulation showed the stronger inhibition of inflammation than the antibiotic. Furthermore, the nanoemulsion reduced the bacterial number to 4.3 × 10 CFU/mL, showing a superior bactericidal effect compared to erythromycin (3.5 × 10^3^ CFU/mL). The histopathological and digital photography of mouse ear tissue determined that both oregano nanoemulsion and erythromycin possessed comparable healing effects. The obtained results established the potential use of the oregano nanoemulsion as an alternative treatment for acne, due to its high antimicrobial and healing effects [58].

Likewise, Mazzarello et al. conducted a non-invasive study on sixty volunteers diagnosticated with mild–moderate acne. The researchers’ aim was to assess the antiacne effect of a galenic formulation containing two essential oils known for their antimicrobial and anti-inflammatory activities (*Origanum vulgare* L. and *Mytrus communis* L. purchased from Bussolengo, Italy) and tretinoin, compared to a commercial product containing clindamycin and tretinoin. The cream used by the tested group contained 3.74% *Myrtus communis* L. essential oil, 0.1% OvEo and 0.025% tretinoin (MOTC), while the positive control group applied the commercial product with 1% clindamycin and 0.025% tretinoin (CTG). Therefore, the MOTC formulation presented a better action in reducing the papular erythema index than CTG after 15 days of treatment. Moreover, MOTC reduced the irritation produced by retinoids in health skin around the acneic lesion with a statistical significance of 0.0329 at 15 days and 0.0017 at 30 days. Both groups were confronted with an increased TEWL due to tretinoin. On the strength of the obtained results, the combination of oregano and myrtle essential oils could be successfully used in order to substitute antibiotics in formulations containing retinoids [111].

Kapustova et al. evaluated the antimicrobial and antibiofilm activity of encapsulated OvEo in poly (ɛ-caprolactone) nanocapsules against several bacteria, including *S. aureus*. The results showed that the encapsulated OvEo presented a more effective antibacterial activity than the pure essential oil, with a MIC value of 0.5 mg/mL and an MBC value of 1 mg/mL. The antibiofilm activity at 0.25 and 0.125 mg/mL of OvEo nanocapsules showed 75% and 43% inhibition, respectively. Therefore, due to the increased antibacterial and antibiofilm activities, nanosystems loaded with oregano essential oil could be a natural alternative treatment for cutaneous disorders associated with bacterial invasion, such as acne [57].

A snapshot of the antiacne activity of OvEo is presented in Figure 7.

## 8. The Wound Healing Effect of *Origanum vulgare* L. Essential Oil

Wounds represent physical, chemical or thermal injuries characterized as a disturbance of the normal anatomy and function of the living tissue. Cutaneous wound healing is a complex process that can easily be influenced by different factors, including bacterial infections, tissue hydration and the presence of inflammation. This healing process is commonly divided into four phases: coagulation (minutes), inflammation (hours → days), proliferation (days → weeks) and remodeling (weeks → months). Due to the fact that the pathology of wounds includes inflammatory processes, oxidative stress and even bacterial infection, natural compounds that encompass efficient anti-inflammatory, antioxidant and antimicrobial activities could be a promising option in wound healing management [112,113,114].

The aim of the study conducted by Avola et al. was to evaluate the anti-inflammatory and antioxidant capacity of OvEo, which was obtained from Esperis S.p.A., Milan, Italy, in the wound healing process on human keratinocytes (NCTC 2544). The main components of OvEo were carvacrol (35.95 ± 0.22%), thymol (25.2 ± 0.27%) and *p*-cymene (21.54 ± 0.35%). The DPPH method assessed the antioxidant activity of OvEo, returning a high value for the SC_50_ parameter (114 ± 6 µg/mL). Subsequently, NCTC 2544 cells were subjected to IFN-γ and histamine to induce ROS generation and treated with OvEo (25 µg/mL). IFN-γ and histamine induced, as expected, high levels of relative fluorescence units, while the cells treated with OvEo showed significantly decreased levels of ROS. OvEo also exhibited decreased levels of inflammation (ICAM-1, COX-2 and iNOS) and matrix metalloproteinase (MMP-1 and MMP-12) biomarkers, reduced the proliferating cell nuclear antigen signal and showed a major protective effect against the formation of DNA adducts in cells stimulated with IFN-γ and histamine. Moreover, advanced cell migration was observed in the injury treated with the essential oil after 72 h using in vitro scratch assay. The reduced biomarkers involved in inflammation and the tissue remodeling effect suggest that *Origanum vulgare* L. volatile oil could be an advantageous plant-based compound in wound healing therapy [115].

In order to overcome the instability of OvEo and to deliver a biocompatible pharmaceutical formulation with applicability in wound treatment, Khan et al. encapsulated the volatile oil in Poly (L-lactide-co-caprolactone) (PLCL) and silk fibroin (SF) through electrospinning. Natural polymers were purchased from China, while the OvEo was provided from San Clemente, CA, USA. PLCL/SF/5% OvEo membrane showed a total inhibition of bacterial growth. In vivo analysis was applied to male Sprague Dawley rats in order to assess the capacity of wound healing. Therefore, the histopathological analysis established that the nanofiber membrane containing 5% OvEo had a high capacity to accelerate wound healing in a qualitative manner. Moreover, H&E and Masson’s trichrome staining procedure demonstrated that the PLCL/SF/5% OvEo membrane improved re-epithelialization, granulation tissue formation, angiogenesis and collagen accumulation. Thus, the findings strongly suggest the possibility of using encapsulated OvEo in nanofibers membrane as a potential wound dressing [116]. One year later, Khan et al. assessed the wound healing effect of a nanofibrous membrane loaded with OvEo (supplied from San Clemente, CA, USA) and zinc oxide as bioactive compounds on diabetic wound induced in male Sprague Dawley rats. Due to the controlled release of the bioactive compounds, the inflammatory response was stopped through MMP-9 biomarker and IL-6 inhibition. The innovative formulation led to the healing of the diabetic wound by enhancing the tissue regeneration, re-epithelialization, vascularization and collagen accumulation. The results demonstrated the capacity of this wound dressing to interrupt the prolonged inflammatory process representative for diabetic wounds [117].

Also, Costa et al. published a systematic review regarding the wound healing effect of thymol and carvacrol. The research group established that the monoterpenes interfere in three stages of wound healing by reducing the inflammatory processes and bacterial invasion, promoting re-epithelialization and tissue remodeling processes and modulating collagen induction. Therefore, thymol and carvacrol are relevant compounds for the expansion of new treatments in wound healing therapy [53].

Other species of *Origanum* were also assessed for their wound healing capacity. Along these lines, Süntar et al. studied the efficacy of a formulation containing an equivalent amount of *Origanum minutiflorum* Schwrd. et Davis and *Origanum majorana* L. essential oils, *Salvia triloba* L. essential oil and *Hypericum perforatum* L. olive oil extract, which were purchased from Antalya, Turkey, by using in vivo models. Therefore, incision and excision wound models were applied to male Sprague–Dawley rats and Swiss albino mice (provided from Ankara, Turkey) in order to assess the wound healing effect of the formulation. As suspected, the crème showed a remarkable wound healing activity on both wound models, with values of tensile strength of 45.2% and 33.3%, respectively, on day 10 of topical application. These results were also supported by the histopathological evaluation. Moreover, the formulation possessed antimicrobial and anticollagenase activities. In conclusion, the obtained results sustain the potential usage of the formulation in acute and chronic wound therapy due to its antimicrobial, anti-inflammatory, re-epithelialization and tissue remodeling effects [118].

A snapshot of the wound healing effect of OvEo is presented in Figure 8.

A snapshot of the cutaneous effects of *Origanum vulgare* L. essential oil is presented in Figure 9.

## 9. Conclusions

The review comprises an up-to-date overview of the studies assessing the pharmacological activities of *Origanum vulgare* L. essential oil on various skin disorders. Therefore, the antioxidant, antimicrobial and anti-inflammatory activities of this essential oil are closely related to the antiaging, antiacne and wound healing effects. Mechanisms of action, such as the inhibition of reactive oxygen species and proinflammatory cytokines, the production of structural and functional damage of the bacteria, antibiofilm activity, anticollagenase and antielastase effects, tissue remodeling and re-epithelialization capacity characterize the cosmeceutical profile of OvEo. Furthermore, several approaches towards the development of modern topical formulations in order to deliver effective and biocompatible preparations have been reported in recent years. Thus, nanosystems loaded with OvEo could be a natural alternative approach to several skin disorders, including skin aging, acne and wounds. This review highlights the great potential of OvEo to successfully enter the dermatological field in order to renew modern treatments.

## Figures and Tables

**Figure 1 antibiotics-11-00549-f001:**
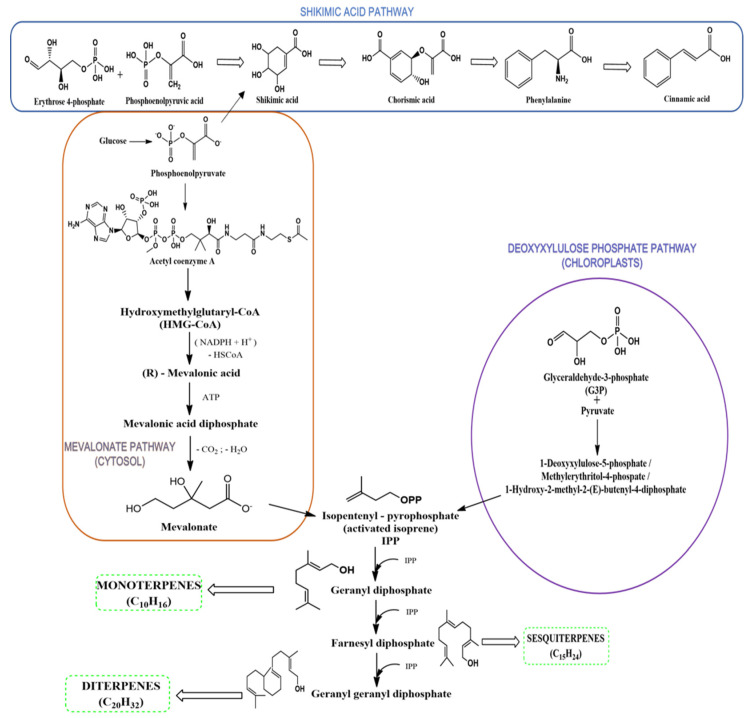
Biosynthesis pathways of terpenoids, according to the literature [24,35,36,37].

**Figure 2 antibiotics-11-00549-f002:**
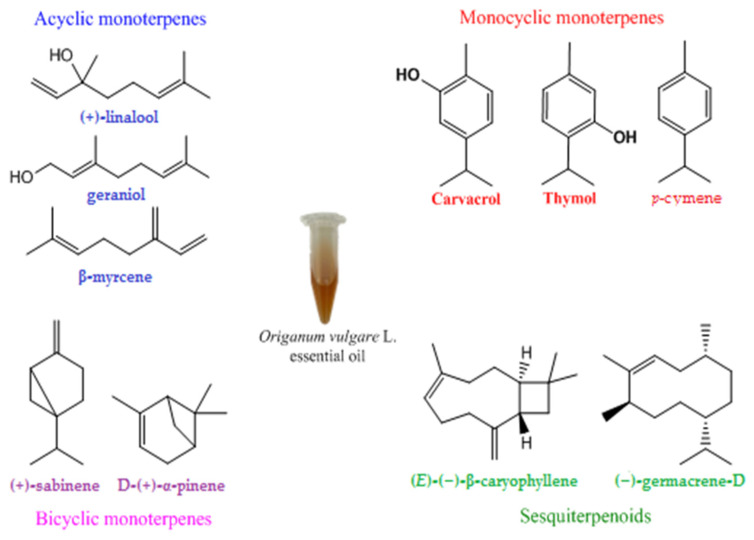
Chemical structures of main constituents of *Origanum vulgare* L. essential oil.

**Figure 3 antibiotics-11-00549-f003:**
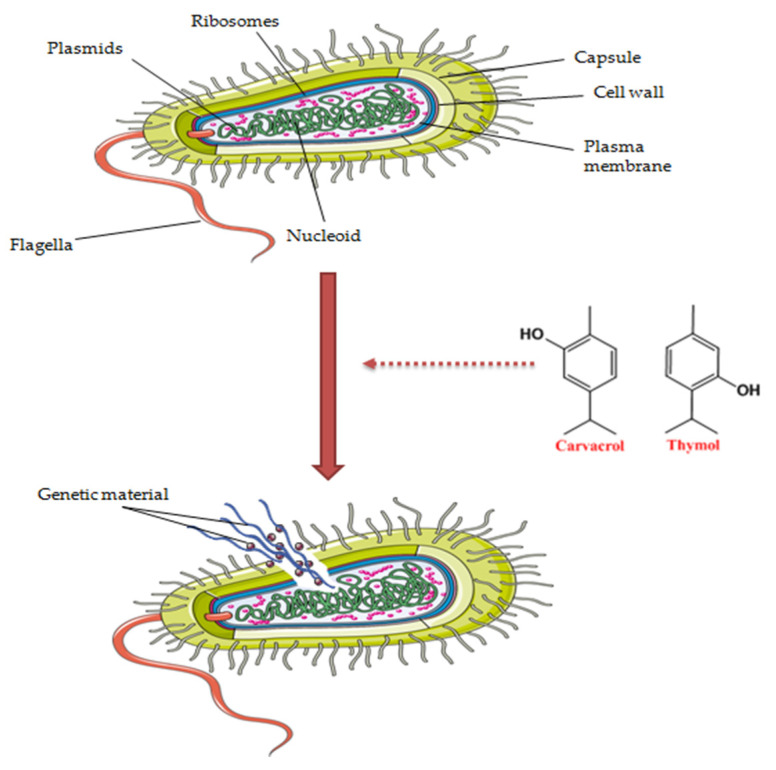
A snapshot of the antimicrobial mechanism of action of OvEo.

**Figure 4 antibiotics-11-00549-f004:**
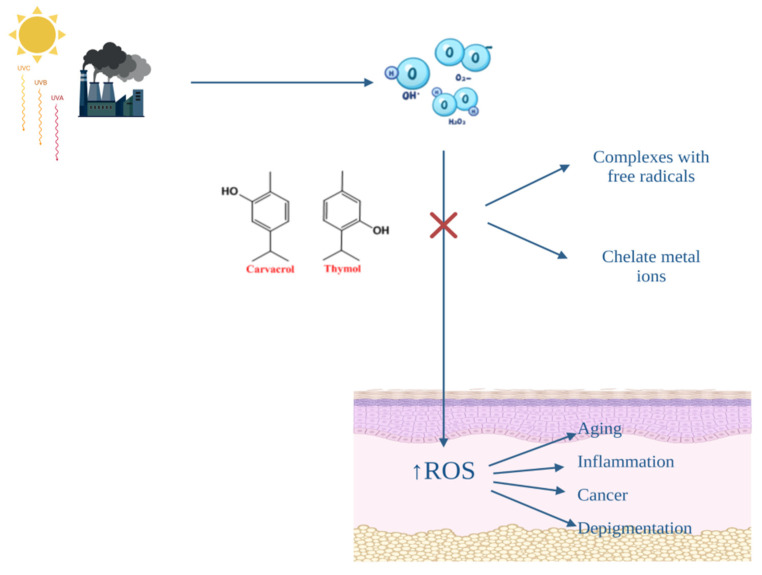
A snapshot of the antioxidant effect of OvEo.

**Figure 5 antibiotics-11-00549-f005:**
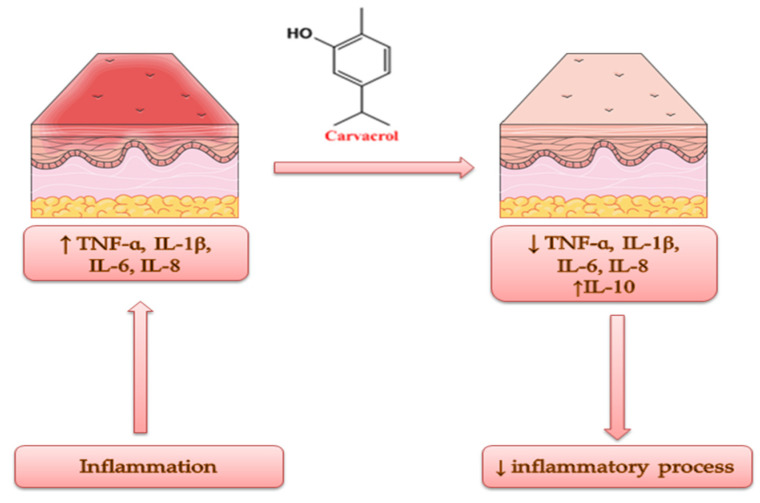
A snapshot of the anti-inflammatory effect of OvEo.

**Figure 6 antibiotics-11-00549-f006:**
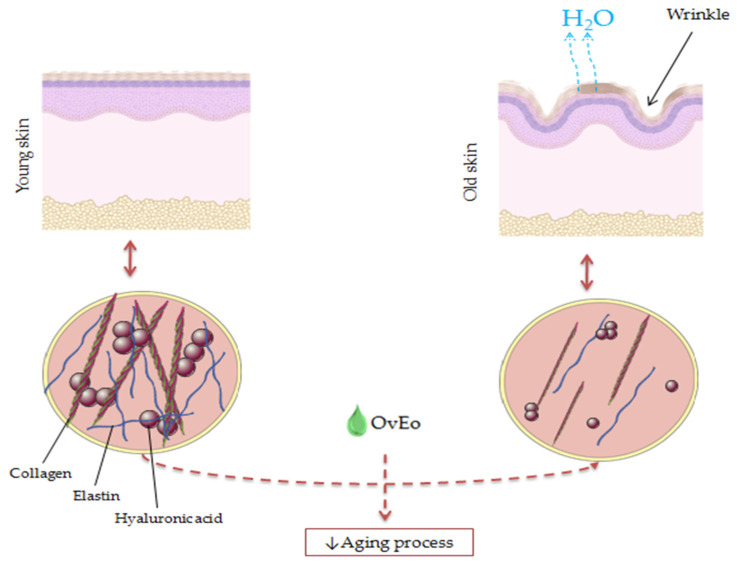
A snapshot of the antiaging effect of OvEo.

**Figure 7 antibiotics-11-00549-f007:**
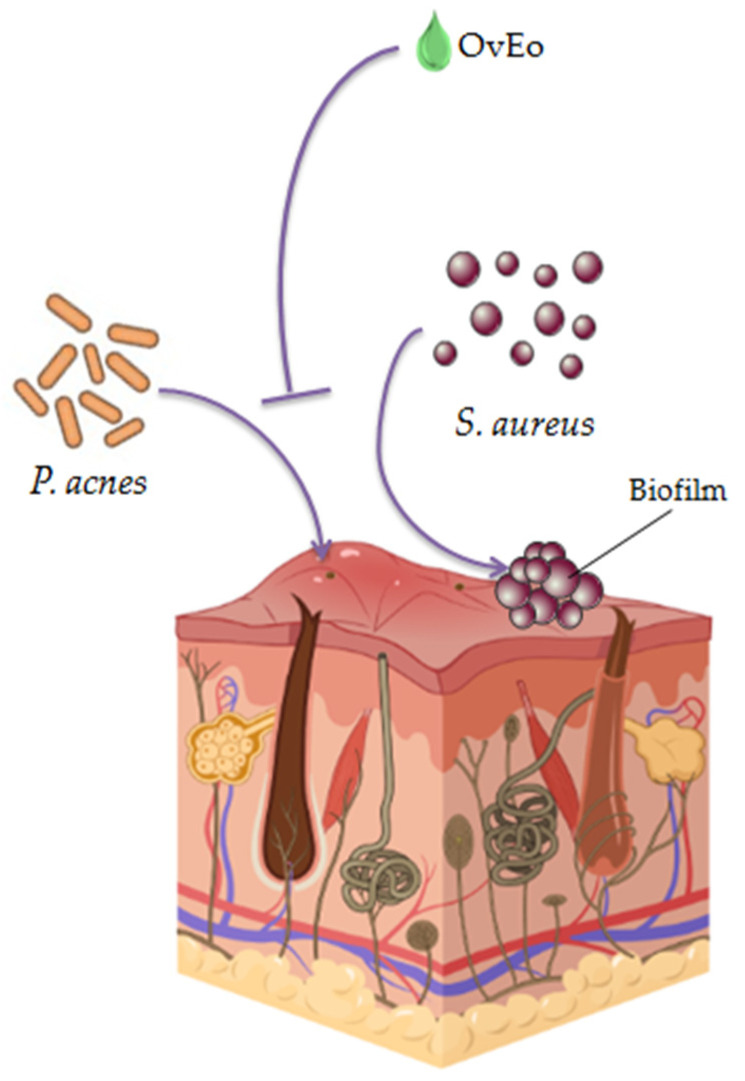
A snapshot of the antiacne effect of OvEo.

**Figure 8 antibiotics-11-00549-f008:**
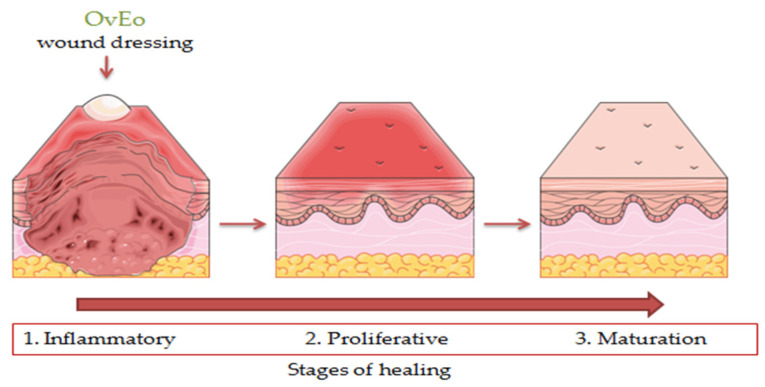
A snapshot of the wound healing effect of OvEo.

**Figure 9 antibiotics-11-00549-f009:**
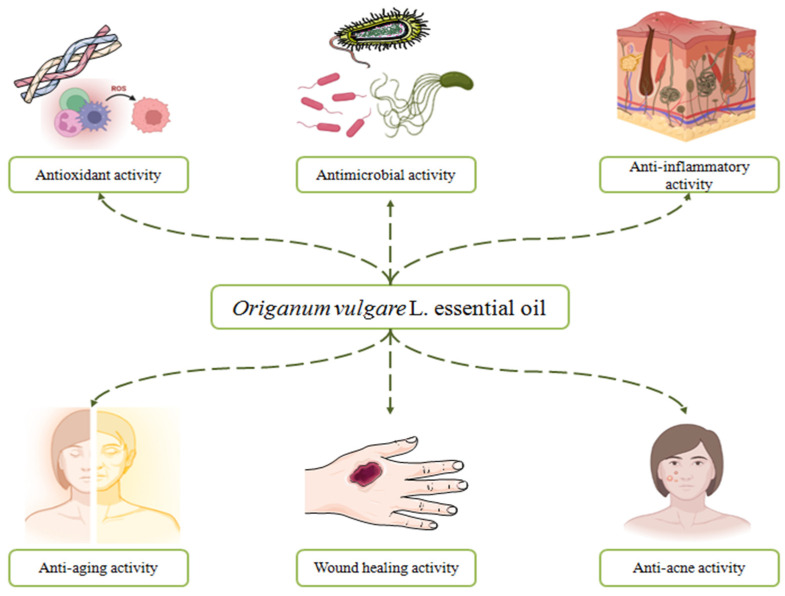
A snapshot of the cutaneous effects of *Origanum vulgare* L. essential oil.

## Data Availability

No new data were created or analyzed in this study.

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
