# Peer review of "An Up-To-Date Review Regarding Cutaneous Benefits of Origanum vulgare L. Essential Oil"

_antibiotics, 2022, doi:10.3390/antibiotics11050549_

Round 1

Reviewer 1 Report

A review article entitled “An up to date regarding cutaneous benefits of Origanum vulgare L. essential oil” has described some health benefits of this medicinal plant.

  • Authors should mention all secondary metabolites isolated form Origanum vulgare L (you can alternatively show them in supplementary file). Please review further research articles to know such metabolites. Also, focus on stereochemistry of molecules. For examples, you did not clearly indicate configuration of metabolites in Figure 1.
  • There is no consistency in the unit showing MIC values (mg; ppm; AU). Please express them in the same unit for example mg. You can convert the reported values accordingly. Line 186: there is a mistake in MIC unit.
  • Figures are very few in your review articles. Please develop graphical representation in each case such as antimicrobial activity, antioxidant activity, anti-inflammatory activity, anti-aging effect, and anti-acne effect. Authors should add other 5-6 figures for a better visibility of this review article.  Please focus to outline mechanism of such biological effects. 
  • Please also mention cytotoxicity effects of Origanum  vulgare L.
  • Authors should reduce assay-type description and suggest to lower the similarity index (which is around 22% in the present form).  

Author Response

Thank you very much for your suggestions. 

Reviewer 2 Report

Title : “An up to date regarding cutaneous benefits of Origanum vulgare L. essential oil”

  1. This reviewer suggests the authors should explain the methodology adopted to make the review, what sources were used for the research, what criteria were used for inclusion & exclusion and also here they should explain / justify the organization of the article, i.e., the sections/subsections included in the review. You can follow some examples with different approaches, that might help you here: https://www.mdpi.com/2073-4395/11/8/1568, https://www.mdpi.com/2227-9717/9/2/223, https://pubmed.ncbi.nlm.nih.gov/32962007/.
  2. After the first mention of species, the authors must use abbreviation, such as Staphylococcus aureus and Origanum vulgare L.
  3. This reviewer understands that if you want to mention the authors like this Guarda et al., the publication year should be included, such as xxxxx et al., 2020.
  4. These two parts “The Benefits of the Antimicrobial Effect on the Skin. The Antimicrobial Activity of Origanum vulgare L. Essential Oil” and “The Anti-Acne Effect of Origanum vulgare L. Essential Oil” should be combine together because they are related story.
  5. This part “The Benefits of the Anti-Inflammatory Effect on the Skin”, i think the authors must concentrate on the effect of Origanum vulgare L. Essential Oil as anti-inflammation on the skin, however, the authors also included articles about the anti-inflammatory activities of OvEo in mouse-airway inflammation or edema. These are indirect evidences.
  6. In general, this reviewer believes that the current study falls beyond the scope of Antibiotics. There is little knowledge on antibacterial activity. Additionally, additional clinical data must be given. The majority of the data in this review are in vitro data.

Author Response

(The authors gave the same response as above.)

Reviewer 3 Report

This is an extensive review on cutaneous effects of Origanum vulgare essential oils. I believe this review adds to the body of literature in the field of phytotherapy. I would like to praise the authors for such an extensive literature review while writing this manuscript. The references are up to date and authors have gathered the most important findings.

I have few suggestions for the authors:

MAJOR - english language needs to be improved

MINOR

1) line 23 - additional space

2) line 62 - you have already introduced EO abbreviation

3) line 63 - drying and storage are separate factors

4) line 101 - additional space

5) figure 1 - reference needed

6) line 125 - tab

7) line 146 - and Escherichia

8) line 147 - and naphthoquinones

9) line 146 - additional space

10) line 360 - was this clinical study?

11) please shortly report limitations of human studies you have mentioned in the manuscript

12) in general, there is no need for extensive reporting of numbers and p values of studies you have included

13) figure 3 - is this your photo/figure

Author Response

(The authors gave the same response as above.)
